# Chronic Inflammatory Status Observed in Patients with Type 2 Diabetes Induces Modulation of Cytochrome P450 Expression and Activity

**DOI:** 10.3390/ijms22094967

**Published:** 2021-05-07

**Authors:** Lucy Darakjian, Malavika Deodhar, Jacques Turgeon, Veronique Michaud

**Affiliations:** 1Tabula Rasa HealthCare, Precision Pharmacotherapy Research and Development Institute, Orlando, FL 32827, USA; ldarakjian@trhc.com (L.D.); mdeodhar@trhc.com (M.D.); jturgeon@trhc.com (J.T.); 2Faculty of Pharmacy, Université de Montréal, Montreal, QC H3C 3J7, Canada

**Keywords:** type 2 diabetes, cytochrome P450, phenoconversion, inflammatory markers

## Abstract

Diabetes mellitus is a metabolic disease that causes a hyperglycemic status which leads, over time, to serious damage to the heart, blood vessels, eyes, kidneys and nerves. The most frequent form of diabetes is type 2 diabetes mellitus (T2DM) which is often part of a metabolic syndrome (hyperglycaemia, hypertension, hypercholesterolemia, abdominal obesity) that usually requires the use of several medications from different drug classes to bring each of these conditions under control. T2DM is associated with an increase in inflammatory markers such as interleukin-6 (IL-6) and the tumor necrosis factor alpha (TNF-α). Higher levels of IL-6 and TNF-α are associated with a downregulation of several drug metabolizing enzymes, especially the cytochrome P450 (P450) isoforms CYP3As and CYP2C19. A decrease in these P450 isoenzymes may lead to unexpected rise in plasma levels of substrates of these enzymes. It could also give rise to a mismatch between the genotypes determined for these enzymes, the predicted phenotypes based on these genotypes and the phenotypes observed clinically. This phenomenon is described as phenoconversion. Phenoconversion typically results from either a disease (such as T2DM) or concomitant administration of medications inducing or inhibiting (including competitive or non-competitive inhibition) a P450 isoenzyme used by other substrates for their elimination. Phenoconversion could have a significant impact on drug effects and genotypic-focused clinical outcomes. As the aging population is exposed to polypharmacy along with inflammatory comorbidities, consideration of phenoconversion related to drug metabolizing enzymes is of importance when applying pharmacogenomic results and establishing personalized and more precise drug regimens.

## 1. Introduction

Diabetes is a complex metabolic disorder affecting the glycemic status of the human body. According to the World Health Organization (WHO), “*Diabetes is a chronic*, *metabolic disease characterized by elevated levels of blood glucose*, *which leads over time to serious damage to the heart*, *blood vessels*, *eyes*, *kidneys and nerves. The most common is type 2 diabetes*, *usually in adults*, *which occurs when the body becomes resistant to insulin or doesn’t make enough insulin*.” [1,2]. Indeed, more than 90% of individuals with diabetes worldwide have type 2 diabetes mellitus (T2DM). The estimated worldwide diabetes prevalence for 2010 was 285 million individuals while, by 2030, the number of individuals with diabetes will rise to 438 million; that is 1 in every 11 adult persons is expected to be diagnosed with T2DM [1,3].

It is generally considered that T2DM is part of a metabolic syndrome (hyperglycemia, hypertension, hypercholesterolemia, abdominal obesity) that usually requires the use of several medications from different drug classes to bring each of these conditions under control (often 8 to 10 drugs; polypharmacy) [4,5]. Many effective oral therapies are available to treat T2DM, including sulfonylureas, meglitinides (i.e., non-sulfonylurea insulin secretagogues), biguanides, thiazolidinediones, α-glucosidase inhibitors, and dipeptidyl peptidase-4 inhibitor [1,6]. However, response to T2DM oral therapies exhibits high inter-subject variability. Indeed, large variability in drug disposition and effects, including glycemic control, or incidence of adverse drug events (ADEs) is observed among patients receiving the same treatment options [7,8,9]. Accordingly, the American Diabetes Association recommends that each oral antidiabetic medication be individualized while taking into consideration drugs’ effectiveness and tolerability [10].

Chronic hyperglycemia related to diabetes is associated with end organ failure. For instance, T2DM is associated with several micro- and macrovascular complications where macrovascular complications such as retinopathy, nephropathy, coronary artery disease, peripheral vascular disease, stroke and neuropathy account for more than 59% of deaths in these patients [11]. T2DM, nonalcoholic fatty liver disease (NAFLD) and cardiovascular disease usually coexist with a prevalence of 70% in T2DM patients. NAFLD increases the incidence of T2DM and expedites the progress of complications later in life [12,13]. Epidemiological studies have shown an increased risk of cardiovascular complications with fasting blood glucose levels of hemoglobin A1C (HbA1c) just above the normal range (7.0%) [14,15]. It has also been shown that achieving specific glycemic goals can substantially reduce the risk of both developing micro- and macrovascular complications and morbidity. In long-term follow-up, a 1% reduction in mean HbA1c was associated with 14% lower risk of myocardial infarction [11,16,17]. Hence, the clinical practice guidelines of the American Diabetes Association recommend that T2DM treatment should aim for an HbA1c ≤ 7.0% [18]. To achieve this goal, pharmacologic treatments (one or more antidiabetic drugs) are ultimately needed in most T2DM patients along with diet and exercise.

Variability in the drug response has been observed in T2DM patients, implying that some T2DM patients appear resistant to some drugs while others are more sensitive to other drugs. Variability in the systemic exposure to medications is a major determinant of interindividual variability in drug response [19]. Inter-subject variability in the metabolism of these drugs is a major factor, among several others, that may influence their pharmacokinetics.

The cytochrome P450 (P450) superfamily is a major enzymatic system that functions mainly as a monooxygenase [20]. P450s are membrane-bound hemoproteins that play an important role in the metabolism of drugs and other xenobiotics [21,22]. Amongst P450s found in humans, CYP1A2, CYP2B6, CYP2D6, CYP2E1, CYP2C9, CYP2C19, CYP3A4 and CYP3A5 are involved in the metabolism (oxidation) of 70–80% of approved drugs [23]. Large interindividual variability in P450 activities has been observed especially for CYP3A (40-fold), CYP2D6 (100-fold), CYP2B6 (50-fold) and CYP2C9 (40-fold) [24,25,26]. Hence, modulation of the expression or activity of P450 enzymes leads to changes in drug disposition which may subsequently affect pharmacodynamic response.

P450 expression and/or activity is controlled by several intrinsic and extrinsic factors such as genetics, environment, concomitant medications, and inflammatory markers [23]. On one hand, phenotypic P450 activity is determined by quantifying specific metabolic clearance pathways following the administration of probe drug substrates. On the other hand, genetic polymorphisms are observed for several P450 isoenzymes, and genetic testing could help predict P450 activity and patients’ phenotypes [27,28]. In some cases, there is a mismatch between the phenotype predicted from genotyping results and the actual phenotype observed in a patient. This phenomenon is called phenoconversion. The majority of P450 metabolizing enzymes can be subjected to phenoconversion due to multiple factors such as the use of certain medications (e.g., inhibitors, inducers), diet, smoking, age, disease conditions, and inflammatory status [29]. Growing evidence suggests that increased exposure to certain proinflammatory cytokines may regulate the expression of certain P450 enzymes leading to a temporary phenoconversion. For example, some studies reported clinical evidence of genotype–phenotype mismatches for CYP2C19 in patients with cancer, hepatitis, and infection associated with severe inflammation [30,31,32,33,34]. Chronic inflammatory conditions observed in patients with cardiac failure, arthritis, renal failure and T2DM could also lead to phenoconversion. Henceforth, phenoconversion may be the explanation behind inconsistent results assessing drug response in pharmacogenetic studies using genetic results, not phenotype, as the primary predictive variable [35].

It is now well recognized that T2DM represents a condition of low chronic inflammation. The increase in some cytokine levels in T2DM patients has been linked to changes in the expression and/or activity of some P450 isoenzymes. In this review, we aim to describe the association between inflammatory factors, T2DM, P450 expression and activity, and potential phenoconversion. Additionally, we provide specific examples of drugs with altered pharmacokinetics in T2DM patients.

## 2. T2DM as an Inflammatory Disease

Multiple hypotheses have been asserted pertaining to underlying causes and pathogenicity of diabetes. T2DM is associated with risk factors that are both unmodifiable (e.g., genetic predisposition, ethnicity, ageing, and history of gestational diabetes) and modifiable (e.g., sedentary lifestyle, obesity, unhealthy diet, physical inactivity, and smoking) [36,37]. Interestingly, there is increasing evidence that several of the modifiable risk factors listed above may activate inflammatory pathways leading to insulin resistance and T2DM [38].

It has been shown that patients with T2DM have increased levels of interleukin-6 (IL-6). This multifunctional cytokine exhibits diabetogenic effects where it stimulates the release of adrenocorticotropic hormone (ACTH) from the pituitary gland which could lead to hyperglycemia [39,40]. Many studies have indicated that the tumor necrosis factor alpha (TNF-α) may affect insulin action thus leading to the development of diabetes [41,42,43]. In animal models, a study by Hotamisligil et al. correlated T2DM to inflammation and demonstrated the role of TNF-α with obesity and insulin resistance [44]. Tuttle et al. and others have shown that both IL-6 and TNF-α levels are significantly increased in diabetic patients compared to non-diabetic patients [45,46]. Increased levels of inflammatory cytokines have also been associated with the progression and complications of T2DM in both human and animal models [38,47,48]. Targeting inflammation, by the identification of pathways that connect inflammation to T2DM, can help treat and prevent T2DM as well as improve risk stratification by using inflammatory markers as potential indexes [49].

Hyperglycemia per se increases levels of IL-6 and is associated with increased risks of T2DM associated morbidity and mortality [50]. In addition to IL-6 and TNF-α, plasma levels of other inflammatory markers including fibrinogen, serum sialic acid, α_1_-glycoprotein acid, plasminogen activator inhibitor, C-reactive protein and cortisol are also reported to be increased in patients with T2DM [51,52,53,54].

Under inflammatory conditions such as T2DM, proinflammatory cytokines are not only produced locally in disease-targeted organs but also can circulate through the body affecting distal organs. Cytokines can alter the activities of numerous transcription factors affecting the regulatory pathways of various proteins including drug metabolizing enzymes. The weight of evidence gathered, both in animal and clinical studies, suggests that the regulation of drug metabolizing proteins, such as P450s, would be interconnected with the T2DM inflammation-related disease. Hence, the disposition of drugs metabolized by various P450 enzymes may be affected by inflammatory conditions such as T2DM. A potential phenoconversion can be expected leading to interindividual differences in drug plasma levels and response.

## 3. P450 Regulation and Inflammation

Drug metabolizing enzymes such as P450s are regulated at transcriptional and post-transcriptional levels [55]. Regulation of P450 mRNA levels by factors such as transcription factors such as the nuclear receptor pregnane X receptor (PXR) and its dimerization with retinoid X receptor (RXR), is responsible for the observed downregulation of P450 enzymes under inflammatory conditions [56]. Furthermore, it has been suggested that transcription factor nuclear factor-kappa B (NF-κB) could control the expression of several P450 enzymes through its interaction with the promotor region of their genes [56] (Figure 1).

Other studies have demonstrated a correlation between P450 isoform expression levels and cytokines levels [56,57,58]. An in vitro study examined the effect of five pro-inflammatory cytokines IL-1β, IL-4, IL-6, TNF-α and interferon gamma (IFN-γ) on the expression of CYP1A2, CYP2C, CYP2E1, CYP3A in primary human hepatocyte cultures [59]. It was shown that IL-1β, IL-6 and TNF-α had a depressant effect on these P450 mRNA levels [60]. Decrease in mRNA levels was restored when cells were treated with CYP inducers. However, when hepatocytes were treated concomitantly with a CYP1A2 inducer (naphthoflavone) and TNF-α, or with a CYP3A4 inducer (rifampin) and IL-6, the magnitude of increase in CYP3A4 mRNA levels was reduced [61]. Similarly, activities of hepatic p450 isoenzymes (cyp1a, cyp2b and cyp2e) were significantly reduced when rats were injected with IL-6, IL-1β or TNF-α [62,63].

In a human study, a single injection of interferon alpha 2b (IFN-α2b) produced a significant decrease (20%) in the metabolism of erythromycin (CYP3As), as determined by the erythromycin breath test [64]. CYP3A4 down-regulation observed in patients with advanced cancer was suggested to be linked to the secretion of IL-6 originating from the tumor (as demonstrated in animal models) [65,66,67]. As P450 enzymes are expressed in various tissues in an isoform selective manner and as they can be selectively regulated by multiple cytokines, P450 isoenzyme expression and activities would be differentially regulated by different inflammatory diseases at different stages of progression.

## 4. Insulin and Glycemic Levels

Hypoglycemia is considered a stressful condition for the human body. Most diabetic patients receiving insulin suffer from episodes of hypoglycemia at least once per week, particularly after having narrow metabolic control in the management of their diabetes. In many studies, hypoglycemia was shown to downregulate cytokine production since cytokine production depends mainly on blood glucose levels [68,69]. According to de Galan et al., insulin-induced hypoglycemia downregulates TNF-α synthesis in a glucose concentration-dependent manner [70]. In that study, insulin effect was measured in non-diabetic and diabetic subjects where it was noticed that in non-diabetic subjects, hypoglycemia downregulated the production capacity of TNF-α. The production capacity of TNF-α was decreased in non-diabetic subjects compared to diabetic subjects at normoglycemia [70].

IL-6 and TNF-α are usually produced by several tissues such as leukocytes, adipocytes, and endothelial cells [71]. In T2DM, it is believed that these cytokines levels are further increased in a hyperglycemia state and that they originate from noncirculating cells, most likely endothelial cells [72]. It is important to note that hyperglycemia-induced oxidative stress leads to the activation of inflammatory gene expression where, according to Guha et al., there is a significant increase in TNF-α arbitrated by reactive oxygen species in cultured monocytic cells incubated with high concentrations of glucose [73].

In one study, the authors suggested that IL-6 and TNF-α show higher concentrations in patients with impaired glucose tolerance compared to those with normal glucose tolerance [72]. A positive association between IL-6 and TNF-α and insulin sensitivity has been reported previously [46], where IL-6 concentrations are negatively correlated with insulin action, while TNF-α is inversely correlated with insulin sensitivity in other studies [74,75].

## 5. P450, Inflammation and T2DM

Clinical trials have demonstrated that multiple factors such as gender, age, diet and drinking habits, concomitant medications and therapy effectiveness, complications from comorbidities, and diseases such as T2DM and diabetic progression, are contributing to variability in the expression and activity of P450s [76,77,78,79]. Changes in P450 expression and function by T2DM appear to be isoform- and tissue-specific [23].

### 5.1. Animal Studies

Protein expressions and activities are altered in animal models of T2DM; however, it must be noted that the way T2DM is induced in these animal models, either chemically, by diet or obesity-induced models, or in genetically engineered animals, could lead to different results on p450 mRNA and protein expression [80]. It is also important to note that chemical induction of T2DM in animals will alter and/or induce p450 pathways and that, although there is high conservation of p450 enzymes amongst species, the extent and catalytic activity between species differ substantially [81]. Nevertheless, a number of studies have reported a modulation of various p450 isoforms in rodent animal models including cyp1a1, cyp1a2, cyp2a1, cyp2b1/2, cyp2c6, cyp2c7, cyp3a1/2 and cyp2e1 expression [82,83,84,85]. A summary of alterations reported is presented in Table 1.

The most common alteration noticed in chemically induced rodent T2DM models is the induction of cyp2e1, where it is induced almost eight-fold in streptozocin (STZ)-induced diabetic rats [86]. This has been explained by different mechanisms. First, increased ketone bodies by post translational stabilization of the enzyme by high ketone levels [87,88]. Second, by insulin secretion which does not only decrease *cyp2e1* gene transcription but also decreases mRNA half-life [89]. This induction can be suppressed by insulin administration, suggesting that insufficient insulin levels could lead to the induction of cyp2e1 [66,82,88,90]. Lastly, regarding decreased growth hormones levels, as T2DM suppresses growth hormone release, this, in turn, can also contribute to cyp2e1 induction through mRNA stabilization or increased transcription [91]. In support of the role of insulin, others have observed that cyp2e1 and cyp2b expression are decreased in primary cultured rat hepatocytes exposed to insulin [92]. In contrast, cyp2e1 mRNA levels and activity were not significantly modulated in a T2DM model from diet-induced obesity in mice [93].

Induction of cyp1a2 was also observed in STZ diabetic rats. Cyp1a2 induction by T2DM resulted in lower verapamil and lidocaine (which are partially metabolized by cyp1a2) area under the curve (AUC) of drug plasma concentrations in STZ diabetic rats [94,95]. In alloxan-induced diabetic rats, induction of cyp1a2 resulted in lower AUC of oltipraz and higher clearance of antipyrine [96,97]. Theophylline is metabolized through cyp1a2 and cyp2e1 to 1,3-dimethyluric acid. In alloxan diabetic rats, it was also shown that T2DM increases the expression of both p450 isoforms resulting in lower theophylline AUC and thus higher 1,3-dimethyluric acid plasma levels [83].

STZ-induced T2DM animal models have been shown to significantly lower the expression of cyp2c11, resulting in decreased metabolism of diclofenac, glibenclamide and nateglinide [98,99,100]. Studies conducted in a genetically engineered T2DM animal models (*db/db* mice) demonstrated that the expression of the hepatic glucocorticoid receptor was increased with associated increased cortisone levels [101]. Higher levels of cortisone resulted in increased levels of cyp2b and cyp2c causing a change in the metabolism of substrates such as phenytoin [100,102]. In contrast, in their diet-induced obesity model of T2DM, Maximos et al. have also shown that cyp2c and cyp2b activity and mRNA expression was downregulated in this mouse model [93,103].

Studies in T2DM animal models have shown both increased and decreased levels of cyp3a expression and activity. On one hand, cyp3a activity was shown to be reduced in diet-induced obesity diabetic animals, while other studies in STZ diabetic rats indicated that it is increased [93,104,105,106]. Consequently, discrepancy on the impact of T2DM on the metabolism of various substrates has been reported. Studies have shown that cyp3a1 expression is decreased in male STZ-diabetic rats but increased in female STZ-diabetic rats [94,107]. It was shown that glibenclamide clearance was reduced in Zucker diabetic fatty (ZDF) rats, suggesting a reduction in cyp3a activity [108]. Maximos et al. demonstrated that cyp3a11 and cyp3a25 mRNA levels were decreased 2- to 14-fold in the liver which was associated with a 21-fold decrease in midazolam metabolism [93]. They also showed that cyp3a modulation was tissue-dependent [93].

The intestine also expresses a variety of p450 enzymes, mainly cyp3a, that contributes to first-pass metabolism. It was found that cyp3a expression was significantly decreased in the intestine of diabetic rats and mice [106,109]. However, insulin was able to partially reverse the decreased in intestinal cyp3a expression in rats [110].

Most studies have indicated that cyp4a expression and activity are induced in T2DM rats. According to Kroetz et al., cyp4a mRNA levels were induced about eight-fold in STZ-induced diabetics rats, which translated into an associated increase in cyp4a protein expression and activity [111]. The molecular mechanism proposed of the induction of cyp4a in diabetic rats is related to the activation of peroxisome proliferator activated receptor-alpha (PPAR-α) increasing *cyp4a* transcription while normal regulation of cyp3a is due to the activation of the pregnane X receptor (PXR), which is known to alter the expression profiles of cyp3a [111].

### 5.2. Human Liver Microsomes and Probes

It has been shown that inflammation-related conditions suppress CYP3A expression and activity in human hepatocytes [57,121]. CYP3A4 expression and activity were significantly lower in hepatic microsomes obtained from diabetic livers. One study reported, using human liver microsomes and midazolam as a probe drug, that CYP3A activity and mRNA and protein expression levels in samples from T2DM patients were decreased 1.6- to 3.3-fold compared to samples from non-diabetic patients [20]. In contrast, CYP2E1 mRNA and protein expression levels as well as activity (using chlorzoxazone as probe drug) were increased significantly [20]. One reason for diabetic induction of CYP2E1 could be hyperketonemia as demonstrated in animal models [122]. CYP2B6 mRNA level has been shown to be reduced under inflammatory conditions in human hepatocytes [57]. Glyburide, a second-generation sulfonylurea, is mainly metabolized by CYP2C9 to cis-3-hydroxycyclohexyl glyburide and trans-4-hydroxycyclohexyl glyburide [123,124]. According to Jain et al. the metabolism of glyburide is decreased in microsomes prepared from the placenta of pregnant women with T2DM compared to that of non-diabetic pregnant women suggesting a decrease in CYP2C9 activity [114].

Together, these P450 enzymes metabolize multiple drugs, and alteration in their enzyme activity or expression could modulate the biotransformation of several drugs in T2DM patients, thus making them either toxic or ineffective (prodrugs).

### 5.3. Clinical Evidence of Phenoconversion in T2DM

Marques et al. studied 17 hypertensive patients, 9 of whom were also diabetic, to measure the pharmacokinetics of the anti-hypertensive agent nisoldipine, a known CYP3A4 substrate [48]. Additionally, lidocaine disposition was assessed as a marker of CYP3A4 activity. These researchers found that nisoldipine clearance was stereospecific in non-diabetic controls, with the (−) enantiomer exhibiting approximately six-fold higher clearance compared to the (+) enantiomer. However, when non-diabetic and T2DM patients were compared, clearances of the (−) and (+) nisoldipine were reduced by 52% and 38% in T2DM patients, respectively [48]. Consequently, AUCs of (−) and (+) nisoldipine were increased by 143% and 94% in T2DM patients compared to non-diabetic controls [48]. The suggested decrease in CYP3A4 activity in T2DM patients was somewhat confirmed by an increase (~21%) in plasma concentration of lidocaine in T2DM patients 15 min after intravenous infusion of the drug [48]. The fact that lidocaine—a relatively high liver extraction ratio drug—was administered intravenously could explain the difference in the magnitude of changes observed between lidocaine and nifedipine although a decrease in the clearance was observed for both drugs. Moises et al. compared the clearance of lidocaine in pregnant woman vs. non-pregnant woman. They also noticed that lidocaine clearance was lower in pregnant women with gestational diabetes, suggesting that gestational diabetes reduced the activity of CYP3A4 (and possibly CYP1A2) which is responsible for the metabolism of this drug although the combinatorial effect of pregnancy and gestational diabetes (i.e., diabetes with a different origin than T2DM), cannot be excluded [115].

Dostalek et al. showed using population pharmacokinetic modelling that T2DM patients had a significantly reduced clearance of atorvastatin lactone when compared to non-diabetic patients [20]. CYP3A4 is responsible for the metabolism of atorvastatin lactone and acid forms into corresponding hydroxylated metabolites with a higher intrinsic metabolic clearance for the lactone form compared to the acid form [125]. CYP3A4 downregulation due to T2DM could result in less atorvastatin lactone clearance thus forcing it to equilibrium with the acid form, which could lead to a higher risk of 3-hydroxy-3-methyl-glutaryl-CoA reductase (HMG-CoA reductase) inhibitor (statin)-induced myotoxicity.

Diabetes is one of leading causes of kidney diseases; consequently, 30–40% of all kidney transplant recipients in the United States are diabetic at the time of transplantation surgery, and an additional 15–20% develop new-onset diabetes after transplant [126]. Cyclosporine is a calcineurin inhibitor used as an immunosuppressant to prevent organ transplant rejection. Cyclosporine is metabolized in both the intestine and the liver by CYP3A4, resulting in the formation of multiple metabolites [127]. Of the most important metabolites of cyclosporine are the 2-hydroxylated metabolites (AM1 and AM9) [59]. Factors that can contribute to its pharmacokinetics (PK) variability include the extent of cyclosporine pre-systemic elimination by CYP3A4 and/or by the efflux transporter ABCB1. Hryniewieck et al. investigated the associations of cardiovascular diseases with cyclosporine metabolites and indices of cyclosporine metabolism [128]. Their results showed lower AM9 metabolite/cyclosporine ratio of their plasma concentrations in patients with diabetes. This finding suggests a decreased hepatic metabolism in T2DM patients.

Tacrolimus, another calcineurin inhibitor used to prevent organ transplant rejection, is metabolized by CYP3A4/5 to active and inactive metabolites [129,130,131]. There is a wide intersubject variability in the tacrolimus dose needed to achieve therapeutic trough levels. Jacobson et al. found that T2DM patients exhibit increased trough levels suggesting a decrease in CYP3A activities [117].

Gravel et al. have shed light on the clinical impact of T2DM on P450s by investigating activities of seven of the major CYPs (CYP1A2, CYP2B6, CYP2C9, CYP2C19, CYP2D6, CYP2E1, and CYP3A4/5) in T2DM compared to non-T2DM patients following the administration of oral probes (caffeine, bupropion, tolbutamide, omeprazole, dextromethorphan, chlorzoxazone, and midazolam) [120]. They concluded that the activity of CYP2C19, CYP2B6, and CYP3A in T2DM patients was decreased by 46%, 45%, and 38%, respectively; these effects were attributed to some cytokines such as IL-6 [120].

Korrapati et al. found that, while a group of T2DM patients did not differ significantly in their theophylline clearance, theophylline metabolism was positively associated with the degree of HbA1c within the diabetic group suggesting that CYP1A2 may be induced by poor glycemic control [118]. Urry et al. and Gravel et al. have demonstrated that an increase in paraxanthine to caffeine ratio is suggestive of an increase in CYP1A2 activity in patients with T2DM and that CYP1A2 activity can be dependent on glycemic levels as well as HbA1c [78,120]. Hong et al. described a clear association between endogenous insulin levels and CYP1A2 activity [132]. The exact mechanism of CYP1A2 modulation by inflammatory factors is not known but studies may suggest the role of aryl hydrocarbon receptor activation in immune responses or under oxidative stress [133].

In regard to other P450 isoforms, CYP2D6 activity was not changed in patients with T2DM using dextromethorphan as a probe substrate [134]. CYP2E1 has been reported to show no change in activity and expression and that it is less sensitive to disease state (e.g., inflammation) or T2DM [23,120,135,136]. In contrast, as mentioned previously, an induction in cyp2e1 has been noticed in STZ-diabetic rats.

## 6. Clopidogrel: A Specific Example of Phenoconversion in T2DM

Previous preclinical and clinical studies have shown that T2DM can be associated with a reduced activity of CYP2C19 [93,120,137,138]. This could result in altered metabolism of substrates metabolized or activated by CYP2C19. Accumulating evidence suggests a sub-optimal response to the antiplatelet agent clopidogrel in patients with T2DM and insulin resistance [119,139]. Clopidogrel, a prodrug, is activated by sequential oxidations mediated by P450s including CYP2C19 and to a lesser extent CYP2B6 and CYP3A4 [140]. In a study with a T2DM-induced obesity (DIO) mouse model it was shown that production of clopidogrel active metabolite was reduced by half in DIO mice compared to lean control animals [141]. Three of the four cyp2c orthologs involved in the production of the active metabolite were downregulated in DIO mice compared to controls. Finally, platelet inhibitory activity of clopidogrel was also shown to be reduced in DIO mice compared to controls [141]. In another preclinical study using wild-type apolipoprotein E (apoE) mice and diabetic apoE deficient mice, it was shown that platelet inhibition by clopidogrel was reduced in diabetic mice compared to wild-types [142].

Angiolillo et al. studied platelet function in patients with T2DM vs. non-diabetic patients taking clopidogrel and aspirin. After 24 h of a clopidogrel loading dose, adenosine diphosphate (ADP)-induced platelet aggregation was significantly higher in patients with T2DM compared to non-diabetic patients. The same effect was noticed after clopidogrel chronic use, where increased platelet aggregation, glycoprotein (GP)IIIb/IIIa activation, and higher P-selection expression were present in patients with T2DM vs. non-diabetic patients, considering patients with T2DM as non-responders [139].

Erlinge et al. have explored the occurrence and mechanism of clopidogrel unresponsiveness in T2DM patients vs. non-diabetic patients in a double-blinded clinical trial [143]. Patients were randomized to receive clopidogrel or prasugrel (a thienopyridine drug that does not require CYP2C19-mediated activation). Results indicated that the clopidogrel T2DM patient group showed an increased proportion of poor responders when compared to the prasugrel group, suggesting that T2DM poor responsiveness was limited to clopidogrel group. Additionally, when comparing T2DM with non-diabetics, clopidogrel active metabolite levels were much lower in the T2DM group [143].

Multiple studies have compared the anti-platelet efficacy of clopidogrel in coronary heart disease (CHD) patients with and without T2DM [119,139]. These studies reported that patients with T2DM comorbidity have reduced anti-platelet efficacy of clopidogrel compared to controls, suggesting some alterations in the activation of the prodrug by CYP2C19 [139,144,145]. A reduction in CYP2C19 activity is associated with worse cardiac outcomes, as demonstrated in the IGNITE trial. In the IGNITE study, Cavallari et al. reported that patients with a loss-of-function allele in the *CYP2C19* gene were at significantly higher risk of myocardial infarction, ischemic stroke, or death within one year following percutaneous coronary intervention, when treated with clopidogrel compared to non T2DM patients. This difference disappeared when patients with loss-of-function alleles were switched to alternate non-CYP2C19 antiplatelet therapy [144,146]. Sweeny et al. conducted a study comparing the platelet inhibition activity of clopidogrel and ticagrelor in T2DM and non-diabetic patients with CHD [144]. They found that ticagrelor performs better than clopidogrel in both T2DM and non-diabetic patients; furthermore, the differences in platelet inhibition between the two groups were considerably less within the ticagrelor group compared to the clopidogrel group. Mechanisms behind CYP2C19 downregulation have been connected to proinflammatory cytokines, where Frye et al. showed an inverse relationship between TNF-α and Il-6 levels and the activity of CYP2C19 in patients with congestive heart failure [147].

As a result of disease-induced phenoconversion, it is possible that the differences between genotypes are less pronounced in T2DM patients. This was demonstrated in a pharmacokinetic study of a novel PPAR-γ modulator R438 [148]. This study was conducted in non-T2DM and T2DM patients for R438, a drug metabolized by CYP2C19 [148]. The differences observed in the disposition of R438 between CYP2C19 poor metabolizers (PMs) and normal metabolizers (NMs) in control patients disappeared in the T2DM patient group, suggesting a phenoconversion of the NM phenotype to a PM phenotype in patients with T2DM.

## 7. Conclusions

T2DM is associated with an increased level of inflammatory markers such as IL-6 and TNF-α. Increased plasma concentrations of IL-6 and TNF-α support the concept that these cytokines have a role in the pathophysiology of T2DM and its complications. Higher levels of these cytokines have been shown to downregulate several drug metabolizing enzymes, especially CYP3As, CYP2B6 and CYP2C19, and it can be suggested that the observed decrease in drug metabolizing enzymes may lead to unexpected rises in drug plasma levels of substrates of these enzymes (drugs discussed in this review are listed in Appendix A). The aging population is exposed to polypharmacy along with inflammatory comorbidities that may affect the expression and activity of drug metabolizing enzymes; both potentially are able to cause phenoconversion. Phenoconversion has a significant impact on drug effects and genotypic-focused clinical outcomes, but also on personalizing therapy in clinical practice.

The literature clearly points to alteration in P450 activity and resultant changes in PK profiles of P450 substrates in patients with T2DM. However, to definitively prove phenoconversion, detailed studies assessing genotype-informed phenotype and observed phenotype will be required in the future. The current state of knowledge provides strong rationale to undertake such trials. These studies will also help interpretation of genetic reports to optimize medication dose and choice in patients with T2DM.

## Figures and Tables

**Figure 1 ijms-22-04967-f001:**
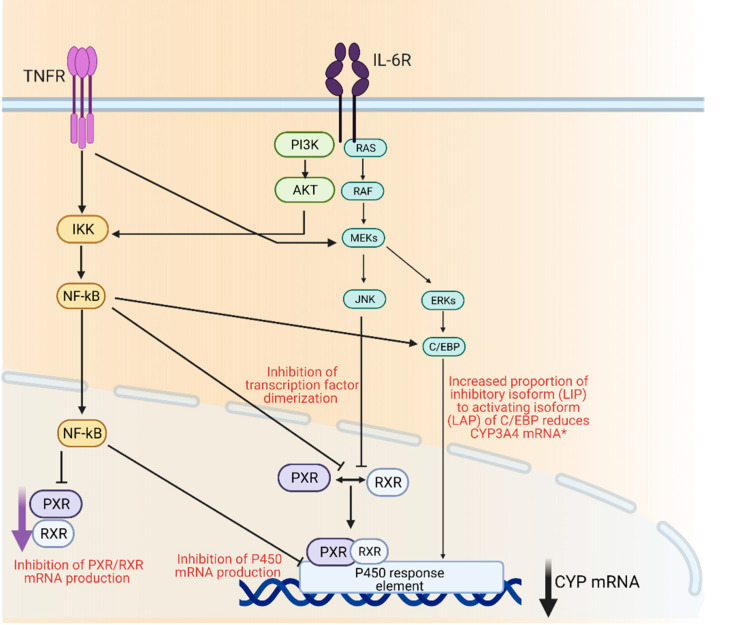
Mechanisms of cytochrome P450 regulation through inflammatory mediators IL-6 and TNFα. AKT; protein kinase B, C/EBP; CCAAT/enhancer-binding protein beta, ERK; extracellular signal regulated kinase, IKK; Iκβ kinase, IL-6R; interleukin 6 receptor, JNK; c-jun N-terminal kinase, MEK; mitogen activated protein kinase, Nf-κβ; nuclear factor kappa light chain enhancer of activate B-cells, PI3K; phosphoinositide 3-kinase, PXR; pregnane X receptor, RAF; rapidly accelerated fibrosarcoma kinase, RAS; small monomeric GTPas, RXR; retinoid X receptor, TNFR; TNFα receptor. Created with BioRender.com.

**Table 1 ijms-22-04967-t001:** Effect of T2DM in multiple models on the expression and activity of several P450 isoforms.

Studies	n	P450s Studied	Effects (mRNA Protein, Activity)	Drug/Probe Level Change	References
**Animals**					
Liver microsomes from STZ-induced diabetic rats	6	1a1	Protein levels increased	-	[85]
		2e1	Protein levels increased 4-fold	-	
		4a1	Protein levels increased 2-fold	-	
Liver microsomes from High-Fat diet (HFD) mice	45 HFD vs. ND	3a11/25	Decreased activity and mRNA levels	Deceased rate of 1-OH-midazolam formation	[93]
		2c	Decreased activity and mRNA levels	Decreased rate of M1-repaglinide and OH-tolbutamide formation	
		2b	Decreased activity and mRNA levels	Decreased rate of OH-bupropion formation	
		2d	No change in activity and mRNA levels	No change in OH-bufuralol formation rate	
Liver microsomes from STZ-induced diabetic rats	12	3a2/1	Protein levels increased 1.5 to 2.5-fold	Verapamil AUC decreased by 22%	[95]
			Protein levels decreased by 67–32%	Decreased AUC_norverapamil_/AUC_verapamil_ by 62%	
STZ-induced diabetic rats	18	1a2	Increased activity	Lidocaine decreased levels 3.4-fold	[94]
			Increased activity	Monoethyl-glycylxylidide (MEGX) increased levels/AUC 2.0 to 2.3-fold	
Alloxan-induced diabetic rats	8	1a2	mRNA levels and expression increased 3.4-fold	Oltipraz decreased AUC by 40%	[97]
		2b1	mRNA levels and expression increased 1.9- fold	Oltipraz decreased AUC by 40%	
		3a1	mRNA levels and expression increased 1.6-fold	Oltipraz decreased AUC by 40%	
Liver microsomes from Alloxan- induced diabetic rats	8	1a2	mRNA levels and expression increased	Omeprazole faster CL_NR_	[112]
Liver microsomes from STZ-induced diabetic rats	4	1a2	Increased protein levels	Antipyrine increased CL 1.5-fold	[96]
Liver microsomes from Alloxan-induced diabetic rats	6	1a2	mRNA levels and expression increased 3-fold	Theophylline decreased AUC by 26.1%	[83]
		2e1	mRNA levels and expression increased 3-fold	Theophylline decreased AUC by 26%	
Liver microsomes from Alloxan-induced diabetic rats	6	1a2	mRNA levels and expression increased 3-fold	1,3-Dimethyluric acid (1,3-DMU) increased AUC by 110%	[83]
Liver microsomes from Alloxan-induced diabetic rats	9	2c11	mRNA levels and expression decreased	Diclofenac slower CL_NR_ 1.3-fold and increased AUC 1.2-fold	[113]
Liver microsomes from STZ-induced diabetic rats	7	2c11	Protein levels and expression decreased by 20%	Glibenclamide increased AUC 5.8-fold	[99]
Alloxan-induced diabetic rats	8	2c6	Protein expression increased	Phenytoin increased CL_NR_	[100]
STZ-induced diabetic rats	5	3a4	Increased expression	Atorvastatin decreased AUC 1.7-fold	[109]
Liver microsomes from STZ-induced diabetic rats	4	2e1	mRNA level and expression increased	-	[85]
Liver microsomes from Zucker diabetic fatty rats	6	4a1/2/3	mRNA and protein levels increased 3-8-fold	-	[111]
**In vitro**					
Primary cultured rat hepatocytes	6	2b	mRNA and protein levels increased 2-fold	-	[92]
		2e1	mRNA and protein levels increased 5-fold	-	
**Human**					
HLM from diabetic patients	6 patients with T2DM vs. no T2DM	3A4	mRNA and protein levels decreased 1.5-fold	-	[20]
		2E1	mRNA and protein levels increased 2.5-fold	-	
HLM from Gestational diabetic patients	9 patients with GDM vs. GDM with Glyburide	2C9	Decreased metabolism	Glyburide (decreased rate of metabolite formation)	[114]
Gestational diabetic patients	6 patients with GDM vs. no GDM	1A2	Decreased expression	Lidocaine increased AUC 1.8-fold	[115]
		3A4	Decreased expression	Lidocaine increased AUC 1.8-fold	
Diabetic patients	9 patients with T2DM vs. no T2DM	3A4	Decreased activity	Nisoldipine decreased CL 1.7-fold	[48]
HLM from diabetic patients	6 liver samples from T2DM vs. no T2DM	3A4	Decreased expression	Atorvastatin lactone decreased CL	[20]
Diabetic patients	7 patients with T2DM vs. no T2DM	3A4	Decreased expression	Cyclosporine decreased AUC	[116]
Diabetic patients	225 kidney transplant T2DM patients vs. no T2DM	3A4/5	Decreased activity	Tacrolimus increased trough levels	[117]
Diabetic patients	8 patients with T2DM vs. no T2DM	1A2	Increased activity	Theophylline increased CL	[118]
Diabetic patients	57 patients with T2DM vs. no T2DM	1A2	Increased activity	Caffeine decreased plasma levels	[78]
Diabetic patients		2C19	Increased expression and activity	Clopidogrel decreased plasma levels	[119]
Diabetic Patients	35 patients with T2DM vs. 38 patients no T2DM	1A2	Increased Activity	C_4 hr_ paraxanthine/C_4 hr_ caffeine	[120]
		2C19	Decreased Activity	Decreased AUC_0-8 hr_ OH-omeprazole/AUC_0-8 hr_ omeprazole 1.9-fold	
		2B6	Decreased Activity	Decreased Ae_0-8 hr_ OH-bupropion/Ae_0-8 hr_ bupropion by 18-fold	
		3A	Decreased Activity	Decreased AUC_0-8 hr_ OH-midazolam/AUC_0-8 hr_ midazolam 1.6-fold	

Abbreviations: Ae; amount excreted unchanged in urine, AUC; area under the curve, C; concentration, CL; clearance; CL_NR_: non-renal clearance, GDM: gestational diabetes mellitus, OH: hydroxy-, T2DM: Type 2 Diabetes Mellitus, ND: normal diet. please note that human P450s are designated CYPXXX whilst animal p450s are designated cypxxx with lower cases characters.

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
