# Peer review of "Chronic Inflammatory Status Observed in Patients with Type 2 Diabetes Induces Modulation of Cytochrome P450 Expression and Activity"

_ijms, 2021, doi:10.3390/ijms22094967_

Round 1

Reviewer 1 Report

In the current paper the authors have undertaken an extensive review on the relationship between inflammatory markers in Diabetes mellitus and CYP450 expression, including impacts on drug metabolism.  The paper is extremely informative, timely and well written.  However, I believe the addition of figures to the manuscript would greatly benefit the reader and add clarity.  For example a schematic figure showing the cellular effects of IL6 and TNF-alpha would be useful (for example in section 2).  Additionally a figure illustrating the mechanisms of P450 regulation would be useful.

Minor editing comments:

Line 48.  Reference citation should read (1,6), not (1) (6)as written

Line 53.  A reference should be added at the end of this sentence.

Line 70.  A reference should be added at the end of this sentence.

Line 75.  Please amend this sentence.  CYP450 act primarily as monooxygenases.  Authors should note not all P450s are monooxygenase – some have other functions such as reductases etc.

Line 77.  The authors should include CYP2E1 as well as CYP1A1 and CYP2B6 as major P450 enzymes

A note of P450 nomenclature should be added to readers not familiar with P450 studies ie human P450s are designated CYP whilst animal P450s are designated cyp.  The authors alternate between both forms throughout the paper and it becomes unclear if they are discussing human enzymes or not.

In some places references are cited with non-italicised ‘et al’ eg line 122, Hotamisligil et al.

Whilst on other places ‘et al’ is italicised eg line 281, Jain et al.  The author should be consistent in formatting.

A second table may be beneficial to the reader.  The table should list the drugs mentioned in the text, its clinical uses, the mechanisms of action and metabolism by P450.

Reviewer 2 Report

In this manuscript, the authors wrote a concise review on the current knowledge of cytochrome P450 phenotype conversion in type 2 diabetes patients. This is of high relevance due to the ever-increasing incidence of T2DM, forming a serious global health problem.

MS is well written although the second half is at some points less clear or confusing, which needs some adjustments

Comments in chronological order:

  1. General comment: although “CYP450” is used by many, the standard abbreviation of cytochrome P450 is: “P450” or “CYP”; correct throughout the MS;
  2. Line 2: On which data/study authors base their affirmation that the inflammatory status of T2DM patients is “low”? This status may fluctuate during the disease. Consider deleting the word “Low” from title
  3. Line 53: reference is missing
  4. Line 76: although Ref 17, by Dr. Ronald W. Estabrook gives an interesting historical overview on the discovery of CYPs and earlier determinations of their role in physiology, a more appropriate reference should be added: e.g. Guengerich FP (2005) Human cytochrome P450 enzymes, in Cytochrome P450: Structure, Mechanism and Biochemistry (Ortiz de Montellano PR ed) pp. 377–530, Plenum Publishers, New York.
  5. Lines 54-67: suggestion: add here a sentence or two on T2DM and NAFLD, e.g. https://www.ncbi.nlm.nih.gov/pmc/articles/PMC6691129/ and https://www.ncbi.nlm.nih.gov/pmc/articles/PMC6063173/
  6. Lines 206-209: although mentioned that T2DM animal models may not represent well the human in vivo situation, I suggest that two issues in this respect should be highlighted: 1) chemical induction of T2DM will alter/induced CYP pathways; 2) although there is high conservation of CYP enzymes amongst species, the extent and catalytic activity between species differs substantially (https://pubmed.ncbi.nlm.nih.gov/17125407/, highlighting that caution should be taken in extrapolation of results to a human situation
  7. Table 1, page 8: cyp isoforms of the primary cultured rat hepatocytes are written with capital letters, which should be in lower case letters, i.e. “2b” and “2e1”;
  8. Line 293: reference (43) is missing.
  9. Line 296: consider deleting this sentence, which is confusing when reading the next sentences.
  10. Line 304: although data of lidocaine intravenously administered “could explain the difference in the degree of changes observed”, this was not the case when observing the effect on two enantiomers (i.e. comparing (-) and (+) nisoldipine increase by 143% and 94% in T2DM patients, compared to non-diabetic controls.
  11. Line 308: conclusion of this study (ref. 109) is incomplete; add: ", although the combinatorial effect of pregnancy and gestational diabetes (i.e. diabetes with a different origin than T2DM), can't be excluded.
  12. Lines 321-335: the use of cyclosporin and tacrolimus in T2DM patients is complex: on one hand the T2DM-incuded inflammation status, causes increased cytokine levels, on the other hand cyclosporin’s and tacrolimus’ immune-suppressor activity has shown to lower principal cytokines (e.g. https://www.ncbi.nlm.nih.gov/pmc/articles/PMC1781759/ and https://pubmed.ncbi.nlm.nih.gov/11407316/): lines 324-325 as well lines 329-330: contradiction in these two cases: diabetes decreases hepatic metabolism but cyclosporin levels are lower? Compare with the case of tacrolimus, lines 334-335, which are in line namely: “T2DM patients exhibit increased trough levels suggesting a decrease in CYP3A activities”
  13. Line 350: “h”?
  14. Lines 350-353: not clear: are there studies that suggest this (in this case references are missing) or studies to be performed confirming these ideas?
  15. Line 358: it’s not clear why clopidrogel is described in a separate subheading, as phenoconversion is described for several therapeutic drug in previous sections; suggestion: use new heading: "6. Clopidrogel: a specific example of Phenoconversion in T2DM"
  16. Lines 370-371: error in sentence: substitute “deficient” with “mice”
  17. Line 380: last part of sentence confusing: which patient group are the non-responders? Substitute "them" with "the former patient group"
  18. Lines 386-388: Unclear sentence, regarding former sentence and given suggestion.
  19. Line 399: substitute “normal” with "non T2DM patients"
  20. Line 400: non-CYP2C19 dependent antiplatelet therapy?
  21. Line 422: Why is CYP2B6 not mentioned here, as a former paper of the authors (ref 114) described: "Mean metabolic activity for CYP2C19, CYP2B6, and CYP3A was decreased in subjects with T2D by about 46%, 45%, and 38%"
